# Genome-Wide Classification and Evolutionary Analysis Reveal Diverged Patterns of Chalcone Isomerase in Plants

**DOI:** 10.3390/biom12070961

**Published:** 2022-07-08

**Authors:** Jianyong Wang, Yifei Jiang, Tong Sun, Chenhao Zhang, Xuhui Liu, Yangsheng Li

**Affiliations:** State Key Laboratory of Hybrid Rice, Key Laboratory for Research and Utilization of Heterosis in Indica Rice, Ministry of Agriculture, College of Life Sciences, Wuhan University, Wuhan 430072, China; wangjianyong90@sina.cn (J.W.); yifeijiang@whu.edu.cn (Y.J.); stong_fx@whu.edu.cn (T.S.); zch_nx@126.com (C.Z.); 2020202040079@whu.edu.cn (X.L.)

**Keywords:** flavonoid, chalcone isomerase (CHI), green plant, gene family, phylogenetic classification, evolutionary dynamics, functional diversification

## Abstract

Flavonoids as a class of important secondary metabolites are widely present in land plants, and chalcone isomerase (CHI) is the key rate-limiting enzyme that participates in catalyzing the stereospecific isomerization of chalcones to yield their corresponding flavanones. However, the phylogenetic dynamics and functional divergence of *CHI* family genes during the evolutionary path of green plants remains poorly understood. Here, a total of 122 *CHI* genes were identified by performing a genome-wide survey of 15 representative green plants from the most ancestral basal plant chlorophyte algae to higher angiosperm plants. Phylogenetic, orthologous groups (OG) classification, and genome structure analysis showed that the *CHI* family genes have evolved into four distinct types (types I–IV) containing eight OGs after gene duplication, and further studies indicated type III *CHIs* consist of three subfamilies (FAP1, FAP2, and FAP3). The phylogeny showed FAP3 CHIs as an ancestral out-group positioned on the outer layers of the main branch, followed by type IV CHIs, which are placed in an evolutionary intermediate between FAP3 CHIs and bona fide CHIs (including type I and type II). The results imply a potential intrinsic evolutionary connection between CHIs existing in the green plants. The amino acid substitutions occurring in several residues have potentially affected the functional divergence between CHI proteins. This is supported by the analysis of transcriptional divergence and cis-acting element analysis. Evolutionary dynamics analyses revealed that the differences in the total number of *CHI* family genes in each plant are primarily attributed to the lineage-specific expansion by natural selective forces. The current studies provide a deeper understanding of the phylogenetic relationships and functional diversification of *CHI* family genes in green plants, which will guide further investigation on molecular characteristics and biological functions of CHIs.

## 1. Introduction

Land plants have evolved a vast array of specialized metabolites, such as a large variety of flavonoids, to cope with diverse biotic and abiotic stressors in the process of colonization and adaptation to terrestrial environments [1,2]. As one of the largest families of (poly)phenols in the world, flavonoids are widely present in plant kingdom, in which they have played an essential role in land colonization and adaptation during the course of plant evolution [2,3]. In the process of plant evolution, it can be inferred that different classes of flavonoids appeared sequentially, accompanied by the emergence of the corresponding genes and enzymes involved in the flavonoid synthesis [4]. According to the early phytochemical studies, chalcones, flavanones, flavonols and flavones first appeared in the ancestors of liverworts and then spread into the entire plant kingdom, whereas the emergence of anthocyanins occurred with the establishment of flowering plants [2,4]. The emergence of isoflavones seems to be a more recent event because they are found mainly in legume species [4]. A previous investigation proposed that UV-B radiation protection and hormone regulation are the original functions of flavonoids in the earliest plants producing flavonoids [5], and their functions have considerably diversified with the subsequent evolution from liverworts to angiosperms [6]. For instance, in extant plants, flavonoids can also function as deterrents or as part of the phytoalexin defense against pathogens and herbivores [7,8], and act as co-pigmentation in tissues (such as flowers and seeds) to attract pollinators [9]. It has also been shown that flavonoids are plant developmental regulators that participate in the mechanisms underlying plant pollen fertility and pollen germination [10]. Furthermore, some of these compounds are also known to have significant pharmacological activity for disease-prevention, with anti-viral, anticancer, anti-allergic, and anti-inflammation activity, and they provide protection against cardiovascular disease, coronary heart disease, obesity, and diabetes [11]. Thus, given their important physiological functions in both plant adaptation and human health, the investigation of flavonoids will be an ongoing focus for research, which could provide a useful insights into how plants evolved the more recent specialized metabolites during the evolution from hydrobiontic plants to land plants.

Flavonoid biosynthesis branches off from the general phenylpropanoid pathway, and chalcone synthase (CHS) and chalcone isomerase (CHI) are the first and second committed enzymes in this pathway [12,13]. As members of the CHI-folding protein superfamily, CHIs include four subclasses, namely, types I–IV [14,15]. Both type I and type II proteins are bona fide CHIs with enzymatic activity catalyzing the stereospecific isomerization of chalcones to yield their corresponding flavanones. Notably, this cyclization reaction can also proceed spontaneously, but greater catalytic efficiency (10^7^-fold) occurs with the aid of the bona fide CHIs [16]. Type I CHIs are widely present in vascular plants and are involved in the production of (*2S*)-naringenin (5,7,4′-trihydroxyflavanone) by exclusively isomerizing naringenin chalcone (4,2′,4′,6′-tetrahydroxychalcone) [17]. However, type II CHIs appear to be specific to legumes and are primarily responsible for the isoflavonoid production, although (*2S*)-naringenin can also be formed in this step [14,17]. According to a recent investigation, type II CHIs are also found in the ancient land plants including liverworts and *Selaginella*, implying that plant type II CHI activity predates the emergence of legume plants [18]. Previous structure analyses of *Medicago sativa* CHI revealed that active site residues and the formation of a hydrogen bond network between active residues are vital for bona fide CHI catalytic activity, and T190 and M191 are thought to determine substrate specificity by lining in the binding pocket [15,19,20]. In contrast to bona fide type I and II CHIs, type III and type IV CHIs are also found in plants but neither of them have CHI activity due to the substitutions of several catalytic core residues, although all types of CHIs in plants possess a similar scaffold conformation observed via structural analysis [13,15]. Type III CHI proteins are widely distributed in both land plants and green algae [2,15], whereas type IV CHIs are thought to be restricted to land plants [2,14,15]. According to a recent study of *Arabidopsis*, type III CHIs belong to fatty acid binding proteins (FAPs) participating in fatty acid metabolism; this type of protein is localized to plastids and can be further subdivided into three subfamilies (FAP1, FAP2, FAP3) [15]. As for type IV CHIs, previous studies have revealed that they could act as enhancers (activators) to contribute to plant flavonoid production [21,22,23]. The underlying mechanism of the type IV CHIs acting as enhancers (activators) in flavonoid biosynthesis is that they could function to bind with bona fide CHIs or CHSs or membrane-bound prenyltransferase (PT1L) and then enhance their catalytic activity; on the other hand, type IV CHIs could also bind and stabilize the ring-open configuration of these products [22,23]. However, recent studies indicate that type IV CHIs appear to be rectifiers rather than activators in plant flavonoid biosynthesis, in which type IV CHIs physically interact with CHSs to achieve the efficient influx of substrates from the general phenylpropanoid pathway to the flavonoid pathway rather than the polyketide pathway [24]. Therefore, the detailed potential mechanisms of type IV CHIs involved in the flavonoid biosynthetic pathway are still complex. Evolutionarily, it is speculated that all CHIs arose from an common ancestral CHI via gene duplication, and bona fide CHIs loosely clustered together with type IV CHIs which probably evolved from a common ancestor type III CHIs, and type III CHIs were thought to be the prototype of the bona fide CHIs [15,25]. The bona fide type I CHIs may be derived from the primitive type II CHIs, which in turn served as an ancestor that was likely retained and fixed in the legume-specific type II CHIs [17,18,26].

Gene duplication occurs widely in plant genomes and is considered as a major driving force to generate the raw genetic material in the course of plant evolution [27,28]. Multi-gene families can be formed following gene duplication and they subsequently undergo four fates accompanied by functional diversification: pseudogenization (loss of function), neofunctionalization (creation of new functions), subfunctionalization (retain partial original functions), and subneofunctionalization (a combination of new functions and original functions) [29,30,31]. These multi-gene duplicates can ultimately be either retained or lost depending on whether a beneficial function is generated during evolution. With the sequencing of plant genomes and transcriptomes, more and more plant *CHI* family genes have been discovered from many species, including primitive seedless plants and higher seed plants. To date, 12 CHIs in *Glycine max* [25], 5 CHIs in *Arabidopsis thaliana* [15], 33 CHIs in cotton [32], 7 CHIs in *Oryza sativa* [33,34], 30 CHIs in six *Citrus* species [35], 31 CHIs in liverwort and *Selaginella* species [18], 56 CHIs in fern species [26], and 11 CHIs in *Dracaena cambodiana* [36], have been identified based on their corresponding genomic or transcriptomic data. Acquisition of CHI enzymatic activity in plants during evolution is an intriguing topic, because both type III and type IV CHIs are devoid of bona fide CHI catalytic activity. Although the evolutionary framework of *CHI* family genes, particularly in the emergence of bona fide CHI catalytic activity in plants, has recently been reported [15,18,26,37], the current phylogenetic classification of *CHI* family genes focuses mainly on the primitive land plants including liverworts, *Selaginellas*, and ferns. Therefore, a more comprehensive and systematic phylogenetic relationship and functional divergence of plant *CHI* gene families is needed to develop a clearer understanding of the evolutionary origins and phylogenetic relationships of plant CHIs from ancestral hydrobiontic plants to higher angiosperm plants. Here, based on the availability of the recently released plant genomes, *CHI* genes were mined by performing a genome-wide survey of 15 representative green plants, with which phylogenetic classification and characteristics were examined. The evolutionary history and functional diversification of the CHI family in plants were systematically explored and discussed based on the comprehensive analysis of phylogeny, genomic structures, selection forces, homologous relationships, cis-acting elements, and expression patterns. The present studies facilitate our understanding of the molecular evolution and functional differentiation of *CHI* gene families in green plants.

## 2. Materials and Methods

### 2.1. Identification and Sequence Features of Putative CHI Gene Families in Green Plants

The genome datasets of 15 representative green plants were retrieved from public databases (Appendix A), with the primary sources being EnsemblPlants (http://plants.ensembl.org/index.html, accessed on 5 December 2021), Phytozome v13.0 (https://phytozome-next.jgi.doe.gov/, accessed on 5 December 2021) and PlantGenIE (https://plantgenie.org/, accessed on 5 December 2021). Pfam domain models of CHI proteins (PF02431, PF16035 and PF16036) were downloaded from the Pfam database ((http://pfam.xfam.org/(, accessed on 11 September 2021). The predicted corresponding putative CHI homologs were retrieved by using a HMMER search against the genome sequence in each plant with a threshold of e-value < 1 × 10^−5^. Some genes would contain multi-transript isoforms because of alternative splicing. Thus, we removed all redundant candidates, and the remaining candidates were subsequently further validated by separately using SMART (http://smart.embl.de/smart/batch.pl, accessed on 21 February 2022) [38], CDD (https://www.ncbi.nlm.nih.gov/Structure/bwrpsb/bwrpsb.cgi, accessed on 21 February 2022) [39], and Pfam (http://pfam.xfam.org/search#tabview=tab1, accessed on 21 February 2022) [40], in normal mode. Sequences with the chalcone domain were ultimately retained for further analysis. The names of putative *CHI* genes identified among these 15 tested species were assigned based on their corresponding chromosomal order. The molecular weights (MW) and theoretical isoelectric points (pI) of all the CHI candidate proteins were calculated by the ExPASy online tool (https://web.expasy.org/protparam/, accessed on 2 March 2022) [41], and the subcellular localizations were predicted by using the WoLFPSORT online server (https://wolfpsort.hgc.jp/, accessed on 5 March 2022) [42].

### 2.2. Multiple Sequence Alignment and Phylogenetic Analysis

Multiple sequence alignments of the above retrieved CHI candidate protein sequences were conducted by using MUSCLE online tool with the default parameters (https://www.ebi.ac.uk/Tools/msa/muscle/, accessed on 26 April 2022) [43]. The final results of multiple sequence alignment were displayed using ESPript 3.0 online tool (https://espript.ibcp.fr/ESPript/cgi-bin/ESPript.cgi, accessed on 28 April 2022) [44]. The best model for accurate maximum likelihood (ML) phylogenetic tree was determined by using ModelFinder software (Australian National University, Canberra, Australia) [45]. The ML tree was subsequently constructed by the IQ-TREE v1.6.3 tool (University of Vienna, Vienna, Austria) with the parameter “-m JTT+F+R5” and 1000 bootstrap replicates [46]. To further support the ML phylogenetic tree, the Bayesian phylogenetic inference was also reconstructed using MrBayes 3.2.6 software (Uppsala University, Uppsala, Sweden) with the parameters of the gamma model and 8,000,000 generations to achieve a convergence diagnostic value <1 × 10^−2^ [47]. The final ML tree and Bayesian tree were visualized using Evolview v2 (PR China University of Chinese Academy of Sciences, Beijing, China.) [48] and FigTree v1.4.3 [49] (University of Edinburgh, Edinburgh, UK), respectively. In addition, the phylogenetic tree of 15 representative green species was built based on the TIMETREE v5.0 online servers (http://www.timetree.org/, accessed on 20 January 2022) [50].

### 2.3. Gene Structure and Chromosome Location

The exon/intron site, length data, and chromosomal location of CHIs were analyzed based on the corresponding genome annotation GFF3 files in each plant. Conserved motifs of all CHIs under study were obtained using the online MEME server with the parameters set at a maximum motif number of 20 and a motif width between 6 and 200 amino acids (https://meme-suite.org/meme/tools/meme, accessed on 16 April 2022) [51]. The final investigation of gene structure was carried out by TBtools v1.09876 (South China Agricultural University, Guangzhou, China) [52]. In addition, the 3D protein structure of all CHIs in this study were created by homology modelling in the online SWISS-MODEL server with default parameters (https://swissmodel.expasy.org/, accessed on 4 March 2022) [53]. The best model result was selected based on the assessment of the highest GMQE values and QMEAN values which demonstrated the reliability of these models. The model and output results was further validated by the assessment with the SAVES v6.0 online server (https://saves.mbi.ucla.edu/, accessed on 10 March 2022).

### 2.4. Anaylsis of Gene Duplication Events, Colinearity, Homology, and Cis-Acting Elements

Gene duplications are thought to be the main reasons for gene family expansions or evolution [54,55]. The MCScanX toolkit (University of Georgia, Athens, GA, USA) was used to analyze the gene duplication events and the putative homologous chromosomal regions [56,57]. Intra- and inter-species genome comparisons were performed using all-vs-all BLAST with a threshold e-value < 1 × 10^−10^ All-vs-all BLAST comparisons of intra-species genomes were used for the identification of gene duplication events in each plant. All-vs-all BLAST comparisons for interspecies genomes were used to assess the genome-wide microcollinearity relationships among different species. Chromosome locations, gene duplication events and collinearity relationships were visualized using Circos software (Canada’s Michael Smith Genome Sciences Center, Vancouver, BC, Canada) [58]. The ratios of nonsynonymous (Ka) and synonymous (Ks) substitutions were used to assess the evolutionary dynamics of duplicate gene pairs using DnaSP v5.0 software (Universitat de Barcelona, Barcelona, Spain) [59]. According to the previously published method, the orthologous groups (OGs) between diverse CHIs in plants were deduced using OrthoFinder v2.0 toolkit (University of Oxford, Oxford, UK) [60]. The Tajima’s D value of all OGs were calculated to assess the selective pressure on evolving OGs using DnaSP v5.0 [59]. The potential cis-acting elements were analyzed by searching the PlantCARE database using the 2 kb upstream region of the start codon of all the *CHIs* in the present study (http://bioinformatics.psb.ugent.be/webtools/plantcare/html/, accessed on 25 January 2022) [61].

### 2.5. Plant Materials and Growth Conditions

Rice ‘RPY geng’ (*O. sativa* ssp. japonica) in our laboratory was used for quantitative real-time transcription polymerase chain reaction (qRT-PCR) analysis. The seeds were firstly germinated in water at 37 °C for 2 days and then were planted in containers with Yoshida solution in the Wuhan University greenhouse, in which the growth conditions were as follows: 14 h light (200 μmol m^−2^ s^−1^, 28 °C)/10 h dark (22 °C) with 65% relative humidity. At 20 days after sowing (DAS), the seedlings were transferred and grown in the field environment with natural summer conditions at the experimental field station of Wuhan University. To investigate the type-specific expression patterns and functional diversification of *OsCHI* genes in various tissues, leaves were separately collected at the four-leaf seedling stage (about two weeks) and at 20 DAS. Shoots were harvested at the four-leaf seedling stage. Inflorescences were separately obtained at the booting and heading stages. Seeds were separately harvested at 7 days after flowering (DAF) and 15 DAF. Embryos and endosperm were collected at 25 DAF, and pistils and anthers were also harvested. All collected samples were immediately plunged in liquid N_2_ and then were stored at −80 °C prior to RNA isolation. Three independent biological replicates (at least 10 plants/replicate) were sampled for RNA extraction.

### 2.6. qRT-PCR Analysis of OsCHI Genes and RNA-Seq Datasets Mining

The expression patterns in various tissues for rice *OsCHIs* genes were examined by qRT-PCR. The specific primers of 7 *OsCHIs* genes for qRT-PCR are listed in Appendix A. Total RNA was extracted (about 0.1 g/each tissue) using TRIzol reagent (Invitrogen, Beijing, China). First-strand cDNAs were synthesized from DNaseI-treated total RNA using M-MLV reverse transcriptase (Promega, Beijing, China) based on the manufacturer’s instructions. The qRT-PCR reaction (10 μL) was performed with triplicate biological samples and two technical replicates in a 96-well plate using the 2 × SYBR Green Master Mix reagent (Bio-Rad, Hercules, CA, USA). The rice *actin1* (LOC_Os03g50885), *eukaryotic elongation factor-1 alpha* (LOC_Os03g08020, *EF-1α*), and ubiquitin-conjugating enzyme E2 (LOC_Os02g42314, UBC) were chosen as controls to normalize the *OsCHIs* expression levels in each sample [62]. The 2^−ΔΔCT^ method was applied to calculate the *OsCHIs* relative expression level. To further investigate whether the type-specific expression patterns and functional diversification of *CHI* genes were common in plants, the RNA-Seq data of various tissues for the target *CHI* genes was extracted from the corresponding publicly available databases for individual plant species: *Triticum aestivum* (http://www.wheat-expression.com/, accessed on 1 May 2022) [63], *Zea mays* (https://www.maizegdb.org/, accessed on 1 May 2022) [64], *Solanum lycopersicum* (http://ted.bti.cornell.edu/, accessed on 1 May 2022) [65], *G.max* (https://www.soybase.org/soyseq/, accessed on 2 May 2022) [66], *Populus trichocarpa* and *Picea abies* (https://plantgenie.org/, accessed on 2 May 2022) [67], and *A.thaliana* (http://bar.utoronto.ca/efp/cgi-bin/efpWeb.cgi, accessed on 3 May 2022) [68]. The expression levels were represented by the values of the fragments per kilobase of transcript per million mapped reads (FPKM). All heatmaps of expression levels in the study were carried out by TBtools [52]. To clearly understand the functional diversification of *CHI* genes in plants, Pearson correlation tests were performed between the *CHIs* in individual plants using the R package.

## 3. Results

### 3.1. Identification of CHI in Green Plants

In order to better understand the evolutionary history of CHI proteins in plants, 15 plant species from different lineages were selected, including one chlorophyte algae, one moss, one lycophyte, one fern, one gymnosperm, and ten angiosperms including three monocots and seven eudicots. All these tested species’ genomes in our study have been sequenced. Among them, *Chlamydomonas reinhardtii* (Cr) is one of the oldest unicellular green plants, *Physcomitrella patens* (Pp) represents the earliest land plants, *Selaginella moellendorffii* (Sm) and *Ceratopteris richardii* (Cri) represent the main lineages of ancient vascular plants, and *P. abies* (Pa) represents the oldest gymnosperms (Figure 1). In all, 122 nonredundant *CHI* genes from these 15 tested species were retrieved by searching the chalcone domain via the HMMER algorithm. At least three members of *CHI* gene family were present in each species, and the number of CHI proteins varied greatly among these tested species. For instance, 18, 12, 10, and 9 CHIs were identified from *T. aestivum* (Ta), *G. max* (Gm), *Z. mays* (Zm), and *Brassica oleracea* (Bo), whereas only three were identified from Cr in the most basal plant lineage in the tree. It should be noted that there were 7, 8, 8, 8 CHI copies in Pp, Sm, Cri, and Pa, respectively, whereas *Cannabis sativa* (Cs) and *Populus trichocarpa* (Pt), belonging to the angiosperms group, contained 6 and 5 copies of *CHI* genes, respectively. In addition, the four remaining tested species in the angiosperm lineages, including *O. sativa* (Os), *A. thaliana* (At), *Chenopodium quinoa* (Cq), and *S. lycopersicum* (Sl), retained 7 *CHI* gene copies. These data indicate gradual trends of expansion or loss in the *CHI* gene family during evolution. More detailed characteristic lists of the putative *CHI* genes for each tested species, including amino acids (aa) number, chromosome locations, molecular weight (MW), and theoretical isoelectric point (pI) etc., are presented in Appendix A. It is noteworthy that the Zm00001d000357 (*ZmCHI10*) gene from *Z. mays* and all the *CHIs* identified from *C. quinoa*, *P. abies* and *S. moellendorffii* could not be mapped to any chromosome conclusively due to the lack of fine genome annotation information.

### 3.2. Phylogenetic Classification of CHI in Green Plants

Previous studies have shown that CHI proteins belonging to the members of CHI-folding protein family can be classified into four canonical types according to their sequence similarity in plants, including types I–IV [14,15]. In this study, 122 CHIs from 15 viridiplantae plants were categorized into four major types based on the maximum likelihood (ML) tree constructed by the IQ-TREE tool (Figure 2 and Appendix A). The established tree consisted of two major branches, in addition to a green algae protein (CrCHI2) from *C. reinhardtii* that was positioned as an out-group to the entire evolutionary origin of CHI proteins in plants. Among them, the type I CHIs included all the vascular species in the present study, which implied type I CHI orthologs are conserved among vascular species and suggests the divergence of type I CHIs occurred before diversification of these vascular species. Intriguingly, three CHI homologs from seedless plants, including two lycophyte homologs (SmCHI3 and SmCHI5) and one fern homolog (CriCHI7), were tightly clustered together with the type I CHIs and placed in the outer layers of the clade, demonstrating that these primitive CHIs may represent an ancient out-group of the higher plant type I CHIs. Within type I CHIs, *P. abies* represented an evolutionary intermediate between seedless plants and angiosperms that diverged after seedless plants and was followed by monocots and eudicots, which is similar to the topological structure of these tested vascular species during evolution (Figure 1). These data show that type I CHIs evolved vertically within vascular species, and the divergence of type I CHIs occurred in the common ancestor of vascular species. The conservation of type I CHI orthologs among vascular species indicates these proteins may have an essential role among these plants. However, only three *CHIs* (*GmCHI4* and *GmCHI10-11* from *G. max*) fell into a clade with type II CHI among the 15 tested plants, which is consistent with the previous finding that type II CHIs are legume specific [13,14,17]. Type III CHIs existed in all viridiplantaes in this study, which indicates that the divergence of viridiplantae plant proteins in this clade may have occurred in the common ancestral species of extant viridiplantae plants, suggesting that these proteins may have an essential physiological role for these plants. Furthermore, type III CHIs could be further subdivided into three clades (consisting of FAP1, FAP2, and FAP3) by clustering with CHI protein homologous to the corresponding counterparts of *Arabidopsis* fatty acid binding proteins (FAPs) [15]. Among them, FAP1 and FAP3 CHIs included members from all viridiplantae plants in the present study, indicating that these genes in this clade had already diversified in a common ancestor of viridiplantaes. CrCHI1 and CrCHI3 from *C. reinhardtii* were ancestral out-groups positioned in the outer layers of the FAP1 and FAP3 clades, respectively. However, FAP2 was found in all land plants, except for *P. abies*. Six proteins, including one *P. patens* homolog (PpCHI6), two *S**. moellendorffii* members (SmCHI4 and SmCHI6) and three fern proteins (CriCHI1, CriCHI4, and CriCHI6), were positioned in outer layers of the clade, suggesting these primitive CHIs may represent an ancestral out-group of FAP2 CHIs of higher angiosperm plants. The absence of FAP2 CHIs in *P. abies* might be the result of gene loss after duplication or incomplete genome-annotation information [69]. Within the type IV CHIs clade, a similar result could be observed, which also covered members from all land plants. These primitive CHIs (such as CriCHI2, SmCHI1, SmCHI7, PpCHI2, and PpCHI7) from the earliest land plants also represented an ancestral out-group of seed plants (including gymnosperms and angiosperms). The conservation of type IV CHIs among land species implies that these clades were already existence in the common ancestor of extant land plants and that the divergence of type IV CHIs had already occurred before these land species diversified. These results suggest type IV CHIs proteins may play an essential physiological role in the growth of these land plants, although their biological function remains to be further elucidated. The above results were robustly supported by the Bayesian–inferred phylogeny tree (Appendix A). It is noteworthy that the results from the topology of the phylogenetic tree showed type III CHIs were clearly separated into two major branches: FAP1 and FAP2 belonged to the smaller branch, while FAP3, clustered together with type I, II and IV CHIs, fell into the other main branch of the tree. Specifically, the whole set of FAP3 CHIs are positioned as the basal layers of the main tree branch, followed by type IV CHIs, type II CHIs, and type I CHIs, indicating that the FAP3 CHIs diverged first, followed by type IV CHIs, which represented an evolutionary intermediate between FAP3 CHIs and bona fide CHIs (Figure 2 and Appendix A). Taken together, these data show that type IV CHIs derived from type III CHIs (FAP3) serve as the ancestors for bona fide CHIs (including type I and type II CHIs), which is consistent with previous findings [15,18,26,37].

To further reveal and quantify the differences between the *CHI* genes in these 15 tested species, an orthologous group (OG) classification was conducted based on the analysis from the OrthoFinder tool [60], which showed that all the *CHI* genes were grouped into eight OGs among the 15 tested species (Figure 2 and Table 1 and Appendix A). The number of OGs was usually five members among these 15 species, except for six OGs in *C. richardii*, *S. lycopersicum*, and Z. *mays*, four OGs in *P. abies* and *S. moellendorffii*, and two OGs in *C. reinhardtii*. In addition, OG1–5 were usually common in all the tested plants, except that *C. reinhardtii* only had OG2 and OG4. There were also no members of OG4 in *C. sativa*, no OG2 in *P. abies*, no OG1 in *P. patens*, and no OG5 in *S. moellendorffii*. However, the three remaining OGs were species-specific, namely, OG6 was found in *P. patens* and *C. richardii*, OG7 was in *C. sativa* and *S. lycopersicum*, and OG8 only existed in *Z. mays*. The classification revealed that the differences in the quantity of *CHI* genes identified in each plant genome were related to gene expansion or loss events of different OGs or types. For instance, the type III (66/122) and OG1–2 (60/122) showed a clear expansion in number compared with other types and OGs: there were 6–11 genes in *T. aestivum* and *G. max*, whereas there were only 2–4 genes in *C. reinhardtii*, *C. sativa*, and *P. trichocarpa*. Finally, we calculated the selection pressure of the eight OGs based on Tajima’s D-values via the DnaSP 5.0 server [59]. The results indicate the Tajima’s D-values of OG1–5 were less than zero (from −0.8743 to −0.7061) (Table 1), indicating that these five OGs were subjected to purification selection during subsequent evolutionary processes.

### 3.3. Analysis of Genome Structure

Earlier studies have shown that gene structure diversity can play an important role in driving the evolution of multigene families [54]. To characterize the potential relationships between the gene structure profiles and evolution history of the plant CHI gene family, the *CHI* gene structures were analyzed according to the established phylogeny tree (Figure 2 and Appendix A). Firstly, a total of 20 putative conserved motifs of 122 CHI proteins were analyzed using MEME suite [51] (Figure 2B and Appendix A). Similar and specific conserved motif arrangements were observed in the two major tree branches; for instance, motifs 3–5 and motifs 7–9 were found in almost all the CHI proteins from the main tree branch, whereas motifs 1–2 and motif 10 were in almost all CHI proteins from the smaller tree branch. Furthermore, the CHI proteins of the same type had a similar motif composition. Motifs 6, 11–12, 14–15, and 17–20 were type-specific motifs. Specifically, motif-17 only existed in type IV, motifs 11–12 were only in bona fide CHIs, FAP2 *CHI* genes contained unique motifs 14–15, and FAP3 *CHI* genes possessed a unique motif 18. These results imply that protein sequences and functions of *CHI* genes between different types varied greatly and have diverged among these 15 tested species during evolution.

Moreover, the intron/exon structures of each OG or type were generated based on the genome GFF3 annotation file to obtain more insights into *CHI**s* structural diversity (Figure 2C). The analysis results indicate that the numbers and lengths of exons or introns tended to vary greatly between phylogeny groups, and even within the same phylogeny groups. The exon numbers ranged from 1 to 12 among these 15 tested species. Among the 28 type I (or OG1) CHIs, four exons (found in 14 CHIs) was the most common, followed by three exons (9 CHIs) and seven exons (2 CHIs). The three remaining *CHIs* (CqCHI5 and PaCHI1-2) retained two, six, and eight exons, respectively. In contrast, all three type II CHIs in the study contained four exons. Similar to the type I CHIs, most of the 24 type IV CHIs contained four exons (found in 13 CHIs), and 5, 2, and 2 CHIs contained 3, 6, and 1 exon(s), respectively. For 66 type III CHIs, 4, 10, and 5 exons were more common in FAP1, FAP2, and FAP3, which contained 17, 19, and 10 *CHIs*, respectively. Consistent with exons, most introns exhibited significant length changes during the CHI evolutionary process, and the intron size also tended to vary greatly, ranging from 0 to 10,800 nucleotides. Notably, type II (or OG1) and type III (or OG4) contained extremely long introns, including *CriCHI1*, *CriCHI5*, *CriCHI6*, *CriCHI8*, and *TaCHI4*. Additionally, we also found there were some *CHI* genes with a single or two introns in 3′UTR and 5′UTR. For instance, *TaCHI14*, *TaCHI13*, *TaCHI12*, *TaCHI15*, *TaCHI18*, *TaCHI17*, *OsCHI2*, *OsCHI4*, *ZmCHI8*, *ZmCHI10*, *CqCHI1*, *SlCHI7*, *PpCHI6*, *CriCHI4*, *CriCHI1*, *PpCHI2*, *CriCHI7*, *PpCHI7*, *CsCHI4*, *AtCHI2*, and *AtCHI7* had a single intron in 5′UTR; *CsCHI2*, *CriCHI6*, and *CqCHI5* contained two introns in 5′UTR; *ZmCHI4* and *ZmCHI9* harbored a single intron in 3′UTR; and *ZmCHI5* retained two introns in 3′UTR. In general, these results suggest that the *CHI* gene function from different OGs or types based on the developed phylogeny have been differentiated by genetic sequence loss or gain events during evolution.

Previous findings revealed that 16 active site residues identified in CHI from *Medicago sativa* were important for bona fide CHI catalysis, including R36, G37, L38, F47, T48, I50, L101, Y106, K109, V110, and N113 as the (2S)-naringenin binding cleft, A49, K97, Y152 as the active site for hydrogen bond network, and T190 and M191 to determine substrate specific binding [15,19,20]. In this study, the polypeptide sequence alignment of the 123 CHIs (including an additional MsCHI as the model) shows that a majority of CHIs from both type I and type II have retained many highly conserved residues for catalytic activity mirrored in MsCHI (although there were some substituted residues in some case). Importantly, the two residues at position 190 and 191 (the residue numbering is based on the MsCHI sequence scheme) postulated to determine the substrate preference were clearly different between type I and type II CHIs. T190 and M191 were only present in three type II CHIs identified from *G. max* (GmCHI4 and GmCHI10-11) and two type I CHIs from *S. moellendorffii* (SmCHI3 and SmCHI5) in the study, whereas both residues were represented by smaller size residues of Ser/Ile (17), Ser/Met (4), Thr/Ile (1), Val/Ile (1), and Phe/Ile (1) in the remaining 24 type I CHIs. Similarly, the residue at site 97 proposed to form the hydrogen bond network also varied between type I and type II CHIs in the present study. This was Lys (K) in the three type II CHIs from *G. max* and different from Met in type I CHIs. In contrast, several catalytic core residues had been substituted in type IV CHIs, except for some residues at positions 47–50 and 106. Likewise, type III CHIs bear substitutions at a nearly complete set of conserved residues positions for CHI catalytic activity, except for the residues at 36, 49, and 101 sites (Appendix A). According to the previous findings, these results reveal that type III and type IV CHIs in the study are devoid of catalytic activity due to the lack of those key core residues, although their catalytic activity needs to be further investigated. Furthermore, 3D-structure models were obtained via homology modeling from the SWIDD-MODEL website (Appendix A and Appendix A). Almost all the CHIs (96/122) were found to belong to four templates, namely, 4doo (47, Chalcone-flavanone isomerase family protein), 4dol (19, *A. thaliana* fatty-acid binding protein At1g53520), 4dok (15, chalcone-flavonone isomerase), and 4doi (15, Chalcone-flavonone isomerase 1), which was in accordance with phylogenetic tree analysis and further strongly supported the reliability of the classification of *CHI* genes. Taken together, these findings from the genome structure analysis were consistent with the phylogenetic inference and classification results discussed above, indicating that the *CHI* gene family has undergone varied functional divergence during the subsequent evolutionary paths in plants.

### 3.4. Chromosome Locations, Homologous Relationship, and Duplication Events

Homologous genes are mainly classified into two types: orthologous and paralogous, and segmental and tandem duplication are the two main processes for gene family expansion or evolution [54,55]. To further investigate the relationships between amplification mechanisms and the evolutionary history of the plant *CHI* homologous gene family, gene duplication patterns and chromosome locations for *CHI* genes from the 14 tested plant species were performed via the MCScanX software [57]. *P. patens* (Pp) was excluded from this analysis because of the incomplete genome-annotation information [69] (Figure 3 and Appendix A and Table 2 and Appendix A). We found that the *CHIs* in each species were unevenly located on the corresponding chromosomes (Chrs). Remarkably, many *CHI* genes were located at the region proximal to the ends or beginnings site of the corresponding Chrs, which were the Chrs dynamic zones and tended to occur as segmental or tandem duplicate events (Figure 3 and Appendix A). Then, 56 genes were found in the genome duplication zone, and 36 duplicate events containing segmental and tandem duplications were identified in these 14 species. Further studies showed that the numbers and modes of duplication events varied greatly among these 14 species, demonstrating that the *CHI* gene amplification mechanisms of these species were different and reflecting a diverse origins for *CHI* genes in the evolutionary course of these plants (Table 2). For instance, duplication events occurred in ten tested species, including *O. sativa* (Os), *T. aestivum* (Ta), *Z. mays* (Zm), *B. oleracea* (Bo), *G. max* (Gm), *S. moellendorffii* (Sm), *C. quinoa* (Cq), *P. patens* (Pp), *P. trichocarpa* (Pt), and *S. lycopersicum* (Sl), whereas no duplication event was identified in *C. reinhardtii* (Cr), *C. richardii* (Cri), *A. thaliana* (At), and *C. sativa* (Cs). Furthermore, segmental duplication events (31) were the major duplication modes in eight tested species, namely, fifteen in Ta, five in Gm, four in Sm, two in Zm and Bo, and one in Os, Pp and Pt, which suggests that segmental duplication played an indispensable role in the gene expansion of these species. In addition, five gene pairs from tandem duplications were also discovered in Bo (one), Gm (two), Cq (one), and Sl (one), indicating tandem duplication also played an important role in the expansion of the CHI family in Bo, Gm, Cq, and Sl. These results suggest that gene duplications played an important role in the amplification of the *CHI* gene families in these tested plant species, especially segmental duplicates.

To elucidate the syntenic conservation and orthologous relationships of *CHIs* across 14 tested plant species, interspecies collinear *CHI* gene pairs were analyzed using comparative genome analysis. A total of 68 collinear *CHI* gene pairs were identified among 14 tested plant species (Figure 3 and Appendix A). These gene pairs were clearly separated into two clusters: monocot-specific and eudicot-specific. For the monocots, 19, 8, and 4 collinear gene pairs were identified between *O. sativa* and *T. aestivum*, between *O. sativa* and *Z. mays*, and between *T. aestivum* and *Z. mays*, respectively. Among them, the six genes *OsCHI2-7* have orthologous relationships with *CHI* genes in both *T. aestivum* and *Z. mays*; the four genes *TaCHI4*, *TaCHI5*, *TaCHI7*, and *TaCHI11* have syntenic relationships with *CHI* genes in both *O. sativa* and *Z. mays*; the two genes *ZmCHI1* and *ZmCHI2* have collinearity relationships with *CHI* genes in both *O. sativa* and *T. aestivum*. No inter-collinear pairs could be observed for *OsCHI1*, *TaCHI1-3*, *TaCHI16*, *ZmCHI4*, *ZmCHI7*, and *ZmCHI10*. The presence of these collinear gene pairs of *CHI* genes among the three monocot species demonstrates that, compared with the relationship between *T. aestivum* and *Z. mays*, there is a closer relationship between *O. sativa* and *T. aestivum*, and *Z. mays*, which is consistent with their evolutionary distance (Figure 1). For the eudicots, 38 collinear *CHI* gene pairs were observed. Among them, there were 7, 2, and 1 collinear gene pairs between *A. thaliana* and *B. oleracea*, *P. trichocarpa*, and *C. sativa*, respectively. In addition, 7, 6, and 3 collinear pairs were separately found between *G. max* and *P. trichocarpa*, *C. sativa*, and *S. lycopersicum*; 2, 1, 1, and 1 gene pairs occurred between *C. quinoa* and *P. trichocarpa*, *G. max*, *S. lycopersicum*, and *C. sativa*; 3, 2, and 2 collinear gene pairs were identified between *P. trichocarpa* and *S. lycopersicum*, between *P. trichocarpa* and *C. sativa*, and between *C. sativa* and *S. lycopersicum*. These findings demonstrate a closer orthologous relationship of *CHIs* between *A. thaliana* and *B. oleracea*, between *G. max* and *C. sativa*, and *P. trichocarpa*, whereas the remaining 18 collinear *CHI* gene pairs showed weak syntenic relationships, such as between *A. thaliana* and *C. sativa*, and between *C. quinoa* and *C. sativa*, etc., which is in accordance with the topological structure of their evolutionary species tree (Figure 1). In addition, no collinearity was observed for the *CHIs* between *A. thaliana* and *G. max*, *C. quinoa*, and *S. lycopersicum*, and between *B. oleracea* and *G. max*, *C. sativa*, *P. trichocarpa*, *C. quinoa*, and *S. lycopersicum*, which may reflect a divergent genomic structure and functional differentiation in the course of these the evolution of these species.

To investigate the evolutionary dynamics of the *CHI* gene family among these 14 tested plant species, the natural selective pressure on evolving genes was evaluated by calculating the ratio of non-synonymous (Ka) and synonymous (Ks) substitution for each duplicated *CHI* gene pair, whereby Ka/Ks > 1, Ka/Ks = 1, and Ka/Ks < 1 represent positive selection, neutral evolution and purifying (or negative) selection, respectively [70]. The results show that Ka/Ks ratios for all duplicate gene pairs were less than 1, ranging from 0.0318 to 0.9420, illustrating that all the identified duplicate pairs underwent strong negative or purifying selective forces during the evolutionary process (Table 2 and Appendix A). This supports the results from the selection tests of orthologous group classification mentioned above (Table 1).

### 3.5. Analysis of Promoters and Expression Divergence

Cis-acting elements present in the promoter region upstream of the gene can regulate gene expression and respond to different environmental conditions by binding to transcription factors, which may provide an important perspective for research on gene functional speculation and differentiation [71]. The potential upstream cis-acting elements of all the *CHIs* among 15 tested species were investigated by searching the PlantCARE database using the 2 kb upstream region of the start codon [61]. A large number of cis-acting elements unevenly distributed on all genes were observed and then were divided into four categories: light response, hormone response, growth and development, and stress response (Figure 4 and Appendix A and Appendix A). The statistics of all cis-acting elements show that the light response category (44% of the all elements) accounted for the largest portion of cis-element occurrence and included 30 types of elements, such as GT1-motif, TCT-motif, Sp1, and GATA-motif, etc. Among them, G-box had the largest frequency (32% of light response elements), followed by the Box 4 element (14% of light response elements). Almost all the genes in the present study contained light response elements in the promoter regions, except for two genes (*PaCHI7* and *SlCHI6*). The second largest category was hormone responses, which accounted for 34% of all elements and included MeJA-responsiveness (CGTCA-motif and TGACG-motif), abscisic acid (ABA) responsiveness (ABRE), gibberellin (GA) responsive (P-box, TATC-box, and GARE-motif), auxin (IAA) responsive (TGA-element, AuxRR-core, TGA-box, and AuxRE), and salicylic acid (SA) responsiveness (TCA-element). ABRE (39% of hormone response elements) was the most common in the hormone response elements, whose portion was higher than the CGTCA-motif (20% of hormone response elements) and the TGACG-motif (20% of hormone response elements). These hormone response elements were ubiquitously present in at least one *CHI* promoter region among these 15 plant species in the study. As for the stress response category, it accounted for 16% of the all elements, the top five main elements were found to be ARE (the anaerobic induction), MBS (an MYB binding site involved in drought-inducibility), LTR (low-temperature responsiveness), GC-motif (anoxic specific inducibility), and TC-rich repeats (defense and stress responsiveness). In the growth and development category (6% of the all elements), these cis-acting elements were distributed in the promoter regions including: the CAT-box responsible for meristem expression, the O_2_-site responsible for zein metabolism regulation, a site responsible for circadian control, an RY-element responsible for seed-specific regulation, and the GCN4_motif responsible for endosperm expression, etc. Taken together, promoters of *CHI* gene families contain light, meristem, drought, anoxic/anaerobic, low-temperature, SA, ABA, GA, IAA, and MeJA response elements, implying that CHIs may play potential roles in plant stress defense, growth and development, and hormone response. Notably, there were some cis-elements preferentially present in individual or several limited genes, such as the DRE element responsible for dehydration, low-temperature, and salt stresses which existed solely in the *SlCHI5* promoter, and the WUN-motif involved in wound response only found in the promoters of *BoCHI9*, *CriCHI6*, *TaCHI15*, and *TaCHI18*. Furthermore, except for the GC-motif responsible for anoxic specific inducibility and the MBS element involved in drought-inducibility, no stress-response element was observed in the promoter regions *of CHI* genes from *C. reinhardtii*. In addition, except for O_2_-site, RY, ARE, MBS, ABA, GA, IAA, and light response elements in the study, no cis-acting element was identified in the promoter of *CHI* genes belonging to OG7. Similarly, no GA, JA, and SA response element was observed in the promoters of *CHI* genes belonging to specific OG6, OG7, and OG7–8, respectively. The results demonstrate that the *CHI* gene families may have undergone or are undergoing functional differentiation during the evolutionary process.

Genes present in the same clade have similar conservative structures or motifs according to the analysis of phylogenetic classification, motif and promoter elements in our research mentioned above, which implies the conserved function in the same clade. Expression patterns can provide important perspectives on the transcriptional divergence of gene families [72,73]. To explore and understand the potential functional diversification between the four types in the CHI family in the path of plant evolution, transcriptional level plant CHIs were analyzed based on the publicly available databases and qRT-PCR. In rice, the type I *OsCHI3* was highly expressed in almost all tested tissues, especially in inflorescences at the heading stage and embryos at 25 DAF (day after flowering), relatively lower in anthers and endosperm at 25 DAF, inflorescences at the booting stage, and seeds at 15 DAF. Compared with *OsCHI3*, four type III *OsCHIs* containing *OsCHI1-2* and *OsCHI4-5* generally demonstrated much lower expression in all tested tissues, except for the anther tissue in which *OsCHI4* had a relatively higher expression level. Interestingly, the remaining two type IV *OsCHIs* (*OsCHI6* and *OsCHI7*) belonging to a gene pair from segmental duplications had similar expression patterns in all tissues. For example, both them were highly expressed in shoots and leaves at the four-leaf stage, and in inflorescences at the heading stage, a relatively lower level in inflorescences at the booting stage, pistils and seeds at 7 DAF, and in leaves at 20 DAS (day after sowing), and were barely expressed in other tissues including anthers, endosperm at 25 DAF, and seeds at 15 DAF. In addition, the correlation coefficient between *OsCHI6* and *OsCHI7* was 0.95 at the transcriptional level (Figure 5A and Appendix A). These investigations were similar to our previous study mining the publicly available rice databases [34]. In *T. aestivum*, there was a clear transcriptional divergence between three types of CHIs, even within the same types. Within type I *TaCHIs*, *TaCHI16* was found to be absent in almost all tissues except for slight expression observed in the spike at the vegetative stage, whereas *TaCHI7*, *TaCHI9* and *TaCHI11* consisting of two duplicated gene pairs were ubiquitously expressed across all tested tissues, except for the slight or absent expression in the grain tissue and the spike at the vegetative stage. As for type III *TaCHIs*, there were 11 type III *TaCHIs*. Among them, the three type III *TaCHIs* including (*TaCH15* and *TaCH17-18*), which included three segmental duplicates, showed similar expression patterns and were transcribed ubiquitously with relatively higher expression level in all tissues, especially in the spike at the vegetative stage. Similarly, *TaCH1-3* (consisting of two duplicated gene pairs) were mainly expressed in the leaves/shoots and in the spike at reproductive stage and were absent from the rest of the tested tissues under study. In contrast, the remaining five type III *TaCHIs* (including *TaCH4-5* and *TaCH12-14*), which consisted of four segmental duplicates, displayed moderate expression levels in the spike tissue but were absent or slightly expressed or had exceptionally high expression in some tissues. Three type IV *TaCHIs* (*TaCHI6*, *TaCHI8* and *TaCHI10*), which included two segmental duplicates, were all highly transcribed in leaves and shoots and the spike at the reproductive stage, relatively lower in the root and grain tissues, and were barely transcribed in the spike at the vegetative stage. Notably, the correlation coefficients between most *CHI* duplicated gene pairs were significantly greater than 0.8 at the transcriptional level, with the exception of three gene pairs including *TaCHI7* and *TaCHI11*, *TaCHI9* and *TaCHI11*, *TaCHI12* and *TaCHI13* (Figure 5B and Appendix A). In *Z.mays*, we first noticed that there were three genes (*ZmCHI4* and *ZmCHI9-10*) barely expressed in any tissues, suggesting these genes had already undergone pseudogenization during the duplication process. The remaining seven *ZmCHIs* had a clear transcriptional differentiation between them, even in the same subgroup. For instance, type I *CHIs* including *ZmCHI1* and *ZmCHI6* (a segmental duplicate gene pair) displayed different expression patterns among various tissues. The former was expressed ubiquitously among all tissues with high expression in the eleventh and thirteenth-leaf at the vegetative stage (V9), seed at 2–10 DAP, and meiotic tassel (V18), whereas the latter was mainly expressed in the eighth-leaf (V9), thirteenth-leaf at reproductive (R2) and flowering (VT) stages. Compared with type I *CHIs*, the four type III *CHIs* (including a duplicated gene pair: *ZmCHI5* and *ZmCHI8*) generally showed much lower expression in all tissues, with the exception of some genes with moderate expression in some tissues. One type IV *CHI* (*ZmCHI3*) was predominantly expressed in silks, and in the eleventh and thirteenth-leaf (V9), with relatively lower expression in seeds, internodes or nodes and root. It should be noted that all *ZmCHIs* were absent or slightly expressed in endosperm (Figure 5C and Appendix A). For *A. thaliana*, no expression of *AtCHI2* was observed in any tissues. The remaining 6 *CHIs* could be subdivided into two main clades based on the expression patterns in various tissues. Among them, *AtCHI3* (type I), *AtCHI5* (type IV), and type III *AtCHIs* (*AtCHI1* and *AtCHI4*) were generally highly expressed throughout all tissues with the exception of some conditions (such as in sliques at the seeds stage (9 and 10 w/o), dry seed, root, and cauline leaf), whereas type I *AtCHIs* (*AtCHI6* and *AtCHI7*) were relatively lower in all tissues. In addition, we noticed the correlation coefficients between *AtCHI6* and *AtCHI7* and between *AtCHI3* and *AtCHI5* were both greater than 0.8 at the transcriptional level (Figure 5D and Appendix A). In *G. max*, the four types of CHIs also showed a clear transcriptional differentiation. For instance, *GmCHI12* (belonging to the type I *GmCHI*) was mainly expressed in flowers, whereas type II *GmCHIs* (*GmCHI4* and *GmCHI10*-*11*) and type IV *GmCHIs* (*GmCHI2* and *GmCHI3*) were predominantly expressed in roots. In contrast, the remaining 6 *GmCHIs* belonging to type III *GmCHIs* were slightly expressed or absent in all tissues. In addition, at the transcriptional level, the correlation coefficients between four duplicated gene pairs were greater than 0.8, including *GmCHI1* and *GmCHI9*, *GmCHI2* and *GmCHI3*, *GmCHI4* and *GmCHI10*, *GmCHI10* and *GmCHI11*, whereas the 3 remaining duplicated gene pairs were less than 0.8 (Figure 5E and Appendix A). In *S. lycopersicum*, a clear transcriptional divergence was also observed between the three types of CHIs, and even within the same type. As for type I *CHIs*, the highest expression of *SlCHI3* was observed in young leaves, followed by moderate expression in vegetative meristems and young flower buds, and *SlCHI2* was barely or slightly expressed in all tissues, although *SlCHI2* and *SlCHI3* belong to a duplicated gene pair. Similarly, within type IV *SlCHIs*, *SlCHI5-7* displayed more widespread expression abundant in all tissues with the expression of some low expression in some conditions, whereas *SlCHI1* was only found to have moderate expression in fruit (Figure 5F and Appendix A). An interesting observation with *CHI* genes in *P. trichocarpa* was that all six *PtCHIs* had more ubiquitous, abundant expression in all tissues, and the Pearson correlation coefficients between these genes were all greater than 0.8 at the transcriptional level, although a higher expression of *PtCHI6* was observed in any tissues (Figure 5G and Appendix A). In *P. abies*, we also observed a clear transcriptional divergence between 8 *PaCHIs* subdivided into three types of CHIs. Among them, *PaCHI1* (type I), *PaCHI4* (type IV), and type III *PaCHIs* (*PaCHI3*, *PaCHI6*, and *PaCHI8*) were expressed ubiquitously throughout all tissues, while the remaining three *PaCHIs* had mostly relatively lower or absent expression in some tissues, although some genes could reach exceptionally high expression in some situations. Notably, the correlation coefficient between *PaCHI1* and *PaCHI4* was observed to be greater than 0.8 at the transcriptional level (Figure 5H and Appendix A). Overall, the expression patterns in various tissues at different developmental stages show that the CHIs in plants present type-specific expression patterns. The majority of the correlation coefficients calculated through Pearson correlation tests between the plant *CHI* genes were less than 0.8, while the correlation coefficients between most *CHI* duplicated gene pairs in *T. aestivum* were significantly greater than 0.8. These data indicate that most different type CHIs in plants have undergone significantly functional divergence, although some functionally redundant or pseudofunctional genes may also be observed.

## 4. Discussion

Flavonoids, as one of the most important secondary metabolites, are widely distributed in the plant kingdom from liverworts to angiosperms, where they may have played an essential role in land colonization and adaptation during the course of plant evolution [2,3]. CHI, as the second rate-limiting enzyme in the flavonoid biosynthetic pathway, has been investigated at biochemical and molecular levels during past decades [2,13,14,15,17,19,20,26,37,74]. However, relatively limited attention had been paid to the evolutionary dynamics and functional divergence of green plant CHIs during evolution. Recently, the availability of the released whole-genomic sequences of various plants has made it possible to undertake comprehensive genome-wide survey and investigate the molecular evolution of the plant CHI family.

Here, 122 *CHI* genes were identified from15 plant species representing a broad range of evolutionary time. The copy number of CHI family proteins varies greatly depending on different plant lineages, ranging from only three copies in the most basal plant, *C. reinhardtii*, to 18 copies in *T. aestivum* (Figure 1 and Appendix A). In this study, seven CHI copies were identified in *A. thaliana*, in contrast to a previous study reporting only five CHI copies [15]. Notably, no expression of *AtCHI2* was observed in any tissues, suggesting *AtCHI2* may be a pseudogene, which is consistent with previous results [15]. In comparison, we identified an extra CHI (PpCHI1) on chromosome 1 of *P. patens*. Similarly, three extra CHIs were identified in the *S. moellendorffii* genome, in contrast to the study by Cheng et al. (2018) which reported five copies of CHIs [18]. These inconsistent results may be due to either the different identification methods or the continuously updated data from the corresponding genome-annotation. We found no direct correlation between the number of genes in a family and the genome sizes. For instance, there were 8 *CHI* genes in *S. moellendorffii* (212 Mbp), and 5 *CHI* genes present in *C. sativa* (877 Mbp). In addition, no significant difference in the number of *CHI* genes was found between *A. thaliana* and *C. quinoa*, although their genome sizes were clearly different, namely, 120 Mbp in *A. thaliana* and 1.32 Gbp in *C. quinoa* (Figure 1). In addition, we discovered 7-8 copies were usually common in these tested plants, and found that the number of *CHI* genes in the early land plants (*P. patens* and *S. moellendorffii*) is greater than that in *C. sativa* and *P. trichocarpa* belonging to angiosperms. These data reveal a gradual trend of expansion or loss of the *CHI* gene family in the path of plant evolution from chlorophyte algae to angiosperms. Previous studies have revealed that duplication events including segment and tandem duplications were the major processes for gene family expansions [54,55]. We found gene duplication events did promote the expansion of the CHI family in some plant species and ultimately led to the difference in the number of *CHI* genes between these tested plants under study. We finally found 56 *CHI* genes present in the genome duplication zone which resulted from 36 duplication events and contained 31 segmental and 5 tandem duplications, indicating the expansion of *CHI* genes among these plants can mainly be attributed to segmental duplication (Figure 3 and Table 2). Notably, the expansion mechanisms of the CHI family were diverse between these tested plants. For example, segmental duplication was specifically observed in *T. aestivum (15)*, *Z. mays*
*(2)*, *S. moellendorffii*
*(4)*, *O. sativa*
*(1)*, *P. patens*
*(1)*, *P. trichocarpa*
*(1)*; segmental and tandem duplications were found in *B. oleracea* and *G. max*; tandem duplication presented specifically in *C. quinoa* and *S. lycopersicum*. These results may reflect the species-specific expansion mechanism of the CHI family during the evolutionary history of these plants. Furthermore, the number of *CHI* genes and their duplication events in *T. aestivum* were far greater than in the other species under study, which may be due to the fact that *T. aestivum* originated from the fusion of three diploid genomes, ultimately leading to the formation of a large number of duplicated genes in genomes with most genes present with at least three functional copies [75]. Interestingly, all the tandem gene duplications only occurred in the type I/II *CHI* genes under study, which is consistent with the results that four bona fide CHIs from *Lotus japonicas* were consist of a tandem cluster in the genome region [17], indicating tandem duplication played an important role in the expansion of the bona fide CHI family. These results reveal that different expansion mechanisms were responsible for the expansion of *CHI* genes in the evolutionary path of green plants. Furthermore, an orthologous group (OG) classification was also used to further investigate differences in the number of *CHI* genes in these 15 tested species (Table 1 and Appendix A). These results show that compared with other OGs, OG1 and OG2 had a clear number expansion phenomenon among these tested plants. For example, the expansion was larger in *T. aestivum* (10) and *G. max* (6), whereas only 2–3 genes were observed in *C. reinhardtii*, *C. sativa*, and *P. trichocarpa*. In addition, species-specific expansion of OGs also promoted the difference in the number of *CHI* genes between species; for example, OG8 was found to only exist in *Z. mays*. Evolutionary dynamic analysis showed all *CHI* homologous genes within this study had been obviously affected by negative or purifying selective forces during evolutionary process (Table 1 and Table 2 and Appendix A), which implies that the current *CHI* gene numbers in plants may be the result of natural selection.

In previous reports, the CHI superfamily was divided into four types of CHI-fold proteins during evolution, namely, type I-IV CHIs [14,15]. Both type I and type II proteins are the bona fide CHIs with enzymatic activity catalyzing the stereospecific isomerization of chalcones to yield their corresponding flavanones. Type I CHIs are widespread in vascular plants and are involved in the production of plant flavonoids [17]. However, type II CHIs appear to be legume-specific and are responsible for the isoflavonoid production [14,17]. In addition, type II CHIs are also found in the ancient land plants including liverworts and *Selaginella*, according to recent investigation [18]. Although all CHIs possess a similar backbone conformation, the type III and type IV CHIs do not have bona fide CHI activity due to the several substitutions of catalytic residues [13,15]. Type III CHI proteins are widely distributed in both land plants and green algae [2,17], whereas type IV CHIs are thought to be restricted to land plants [2,14,15]. Type III CHIs belong to the fatty-acid-binding proteins (FAPs) participating in fatty acid metabolism, which are thus subdivided into three subfamilies (FAP1, FAP2, FAP3), according to a recent study from *Arabidopsis* [15]. As for type IV CHIs, recent studies have revealed that they could act as enhancers (activators) or rectifiers to play indispensable roles in plant flavonoid biosynthesis [21,22,23,24]. The present study reveals that 122 CHI proteins from 15 green plants were subdivided into four major subfamilies based on the genome structure, motifs, phylogenetic relationship from maximum likelihood (ML) and Bayesian tree analyses. Type III CHIs were further subdivided into three clades including FAP1, FAP2, and FAP3 (Figure 2 and Appendix A). Type I CHIs included all the vascular species in the present study. At the same time, three CHI homologs from seedless plants including two lycophyte homologs and one fern homolog, were tightly clustered together with the type I CHIs, demonstrating that the divergence of type I CHIs predated the emergence of vascular plants. In addition, based on the position of these tested vascular species in the evolutionary path (Figure 1), we found the type I CHIs have evolved vertically within vascular species: lycophyte plant (*S**. moellendorffii*) diverged firstly, followed by fern plant (*C. reinhardtii*) and gymnospermae (*P. abies*), and the latest to diverge were monocots and eudicots. However, only three CHIs from *G. max* belonged to type II CHI, indicating that type II CHIs seem to be legume-specific [14,17]. Notably, we found two CHIs (SmCHI3 and SmCHI5) from *S. moellendorffii* clustered together with type I CHIs, in contrast to the study by Cheng et al. (2018), in which they reported only one SmCHI copy clustered together with type II CHIs [18]. We infer that these inconsistent results may be due to the different methods applied in constructing the phylogenic tree and identifying CHIs. Interestingly, according to previous results, the bona fide SmCHI exhibited more favorable activity to naringenin chalcone, which was similar to the type I CHI enzyme characteristic [18,26], whereas T190 and M191 were present in the two CHIs (SmCHI3 and SmCHI5) under study similar to the type II CHI feature (Appendix A). These results hint that the two CHI enzymes (SmCHI3 and SmCHI5) from *S. moellendorffii* likely served as an evolutionary intermediate between type II and type I enzymes in the evolutionary path to vascular higher plants [18,26]. Type III CHIs were found to be present in all green plants in this study, including the unicellular algae, which implies that the fatty acid metabolism pathway was established in the common ancestral species. How chlorophyte algae evolved to terrestrial plants is still an interesting topic [76,77]. Our investigation showed that *C. reinhardtii* possessed two type III CHIs as an ancestral out-group of this clade, similar to land species. On the other hand, type III CHIs mediated regulation of fatty acid metabolism pathways [15], which may have helped plants adapt to the various land conditions. Thus, the diversification of the type III CHI family predating the emergence of land plants may provide insights into land plant evolution. As expected, the type IV CHIs clade included members from all land plants in the study, which reveals that the divergence of type IV CHIs occurred prior to the emergence of land plants and after the chlorophyte algae split. Based on recent reports, we propose that type IV CHIs served as activators or rectifiers for the plant flavonoid biosynthesis [21,22,23,24], helping the land plants adapt to various land environments during evolution. Notably, the whole set of FAP3 CHIs in the study, tightly clustered together with a full set of type IV CHIs, type II CHIs, and type I CHIs, fell into the main branch of the tree. Specifically, FAP3 CHIs was an ancestral out-group placed in the outer layers of the clade, followed by type IV CHIs, type II CHIs, and type I CHIs (Figure 2 and Appendix A). According to our analysis, we propose that a common ancestor of CHI was present in early plant ancestors. Subsequently, multi-protein families of CHIs formed by duplication events during evolution from hydrobiontic plants to land plants. The bona fide CHIs, as an adaptive process of land plant species, are derived from type IV CHIs that evolved from the common ancestor FAP3 (type III CHIs) by undergoing continual structural evolution, which is similar to previous findings [15,18,26,37].

To investigate the evolutionary implications of the CHI family in the evolutionary process of green plants, it is best to cross reference the relationship between CHIs phylogeny and their functions. Functional diversification after gene expansion could occur at the transcriptional level. To date, 12 differentially expressed CHIs in *G. max* [25], 5 tissue-specific CHIs in *A. thaliana* [15], 33 CHIs in cotton [32], 7 CHIs in rice [33,34], 30 CHIs in six *Citrus* species [35], 31 CHIs in liverwort and *Selaginella* species [18], 56 CHIs in fern species [26], and 11 CHIs in *D. cambodiana* [36], have been reported. These observations showed that CHIs usually have tissue-specific expression in various tissues, and often respond to various environmental stressors. Intriguingly, in all cases, type I-IV CHIs showed clear expression diversity between them, which corroborated our transcriptional results that CHIs in eight seed plants (except for in *P. trichocarpa*) showed type-specific expression patterns among various tissues. This is further supported by the finding that most of the correlation coefficients between *CHI* genes in each plant were less than 0.8 at the transcriptional level (Figure 5 and Appendix A). Promoter analysis showed most *cis*-acting elements preferentially exist in the promoter region of individual or some type *CHI* genes in each plant for stress response (such as DRE, WUN-motif, and low-temperature) or hormone response (except for ABA) (Figure 4 and Appendix A). These results support that multi-type CHIs have the potential to participate in diverse regulation during the process of plant growth and development. Furthermore, different members in the same type were found to possess specific and/or redundant functions. For instance, the correlation coefficients of most *CHI* duplicated gene pairs (28/36) were significantly greater than 0.8 at the transcriptional level, implying the functional redundancy between them, with the exception of eight gene pairs: *SlCHI2* and *SlCHI3*, *ZmCHI1* and *ZmCHI6*, *TaCHI7* and *TaCHI11*, *TaCHI9* and *TaCHI11*, *TaCHI12* and *TaCHI13*, *GmCHI5* and *GmCHI7*, *GmCHI6* and *GmCHI8*, *GmCHI11* and *GmCHI12*. Interestingly, we also observede that some *CHI* genes located in different branches may also show functional redundancy or supplement other genes, as indicated by the correlation coefficients between *PaCHI1* (type I) and *PaCHI4* (type IV) and between *AtCHI3* (type I) and *AtCHI5* (type IV), which were observed to be greater than 0.8 at the transcriptional level. In addition, some pseudogenes may also be found in this study, such as *AtCHI2* [15], *ZmCHI6*, *ZmCHI9*, and *ZmCHI10*. These findings were similar with the theoretical models of gene duplication: pseudogenization, neofunctionalization, subfunctionalization, and subneofunctionalization [29,30,31]. Therefore, the functions of CHIs are complex and their functions appear not to be simply classified based on the phylogenetic relationship. Despite the lack of the potential one-to-one relationships between CHIs phylogeny and their function, our data still can provide perspectives to guide future research into CHI functions, particularly in non-model plants. For example, we found the correlation coefficients between *PaCHI1* (type I) and *PaCHI4* (type IV) and between *AtCHI3* (type I) and *AtCHI5* (type IV) were both greater than 0.8 at the transcriptional level. Recent studies have shown that *AtCHI5* as an activator can function with bona fide *AtCHI3* to promote plant flavonoid production [22]. Thereby, we inferred that *PaCHI4* may also function as an activator contributing to the flavonoid production in *P. abies*, although the underlying mechanisms still need further research. Taken together, by combining this promoter analysis and expression profiling data, we propose most CHIs in plants have undergone significant functional divergence or are undergoing functional differentiation during the evolutionary process via gene duplication, although some functional redundant or pseudofunctional genes may also be present. An evolutionary explanation for these results may be that the complex functions of CHIs in plant development are likely derived from the natural selection pressure in the evolutionary path of green plants, and ultimately also have in turn contributed to plants’ adaptability in nature.

## 5. Conclusions

In summary, in the present study we examined the phylogenetic classification and characteristics of 122 *CHI* genes among 15 green plants from chlorophyte algae to angiosperms. Diversifications of these *CHI* genes were observed based on a genome-wide survey and analyses of phylogenetic relationships, genome structure, duplication events, evolutionary dynamics, cis-acting elements, and expression patterns. These results support the notion that a common ancestor of CHI located in early plant ancestors, the bona fide CHIs, are derived from type IV CHIs which evolved from the common ancestor FAP3 (type III CHIs) by undergoing continual structural evolution and gene duplication from the natural selection pressure in the evolutionary path of green plants. The CHI functional divergence has been potentially affected by the amino acid substitutions at the key core residues due to natural selection pressure during evolution. These results are further supported from the observation of promoter elements and transcriptional divergence between *CHI* genes. Taken together, the findings of this study deepen and enrich our understanding of the molecular characteristics and biological functions of various CHIs in green plants at the evolutionary level.

## Figures and Tables

**Figure 1 biomolecules-12-00961-f001:**
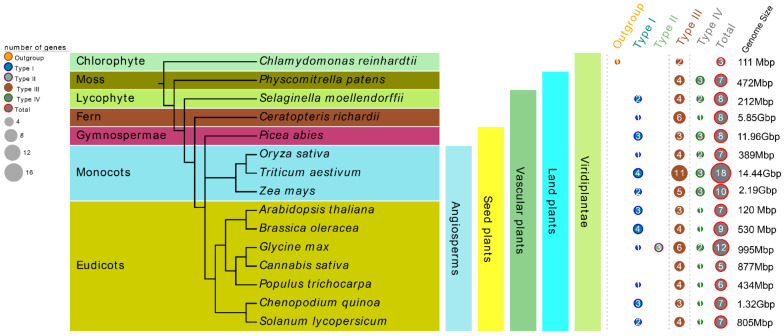
Comparison of the distribution of CHIs among fifteen selected green species representing a broad range evolutionary time. The CHI type class was classified according to the phylogeny analysis.

**Figure 2 biomolecules-12-00961-f002:**
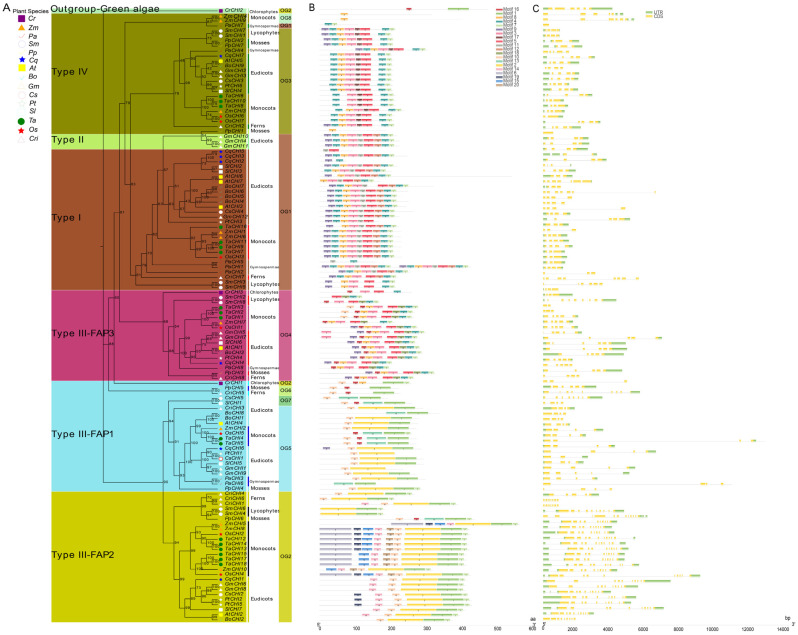
Phylogenetic relationships and genomic structures of green plant *CHI* genes. (**A**) A phylogenetic tree of plant *CHI* genes was built displaying the evolutionary history of these genes. The maximum likelihood (ML) phylogenetic tree was developed using the IQ-TREE tool with 1000 bootstrap replicates. Details of classifications and orthologous groups (OGs) are shown in different colors. Os, Zm, Ta, At, Bo, Cq, Cr, Cs, Gm, Pp, Pt, Sl, Sm, and Cri represent *Oryza sativa*, *Zea mays*, *Triticum aestivum, Arabidopsis thaliana*, *Brassica oleracea*, *Chenopodium quinoa*, *Chlamydomonas reinhardtii*, *Cannabis sativa*, *Glycine max*, *Physcomitrella patens*, *Populus trichocarpa*, *Solanum lycopersicum*, *Selaginella moellendorffii* and *Ceratopteris richardii*, respectively. (**B**) Motif compositions of plant *CHI* genes. Differently colored boxes represent different motifs, and the numbers in each box indicates the order of motifs. (**C**) The structure of exon/intron and untranslated regions (UTRs). The green and yellow boxes and grey lines represent exons, UTR and introns, respectively. FAP—fatty acid binding protein.

**Figure 3 biomolecules-12-00961-f003:**
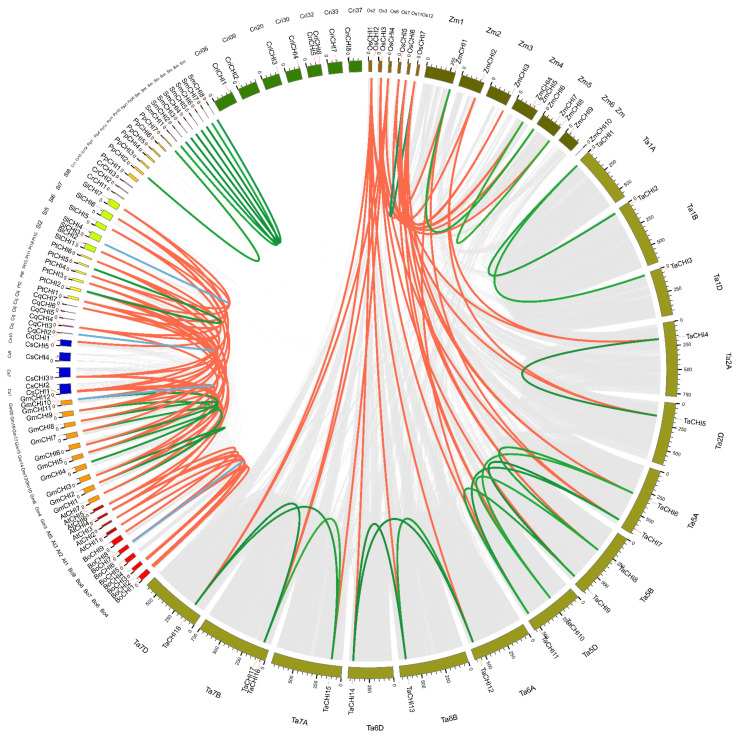
Extensive microcollinearity of *CHI* gene pairs across green plants. The circle plot was created by Circos software. The chromosomes in each plant are indicated by different colors. Os, Zm, Ta, At, Bo, Cq, Cr, Cs, Gm, Pp, Pt, Sl, Sm, and Cri represent *O. sativa*, *Z. mays*, *T. aestivum, A. thaliana*, *B. oleracea*, *C. quinoa*, *C. reinhardtii*, *C. sativa*, *G. max*, *P. patens*, *P. trichocarpa*, *S. lycopersicum*, *S. moellendorffii* and *C. richardi*, respectively. Blue and green curved lines separately represent tandem and segmental duplication events, and red curved lines indicated the syntenic relationships between inter-species. Only the chromosomes containing *CHI* genes were used in the analysis. The scale along each chromosome is in mega base pairs (Mbp).

**Figure 4 biomolecules-12-00961-f004:**
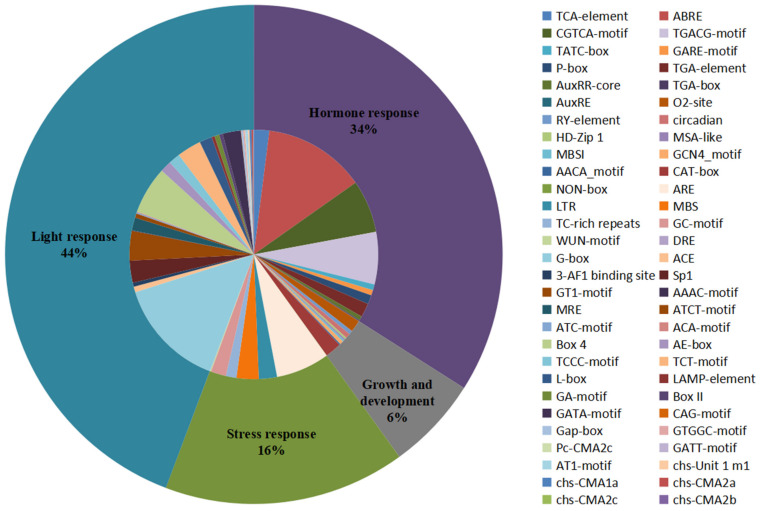
Cis-acting elements identified in all *CHI* genes in 15 representative green plants. Pie charts of different sizes indicated the ratio of each promoter element in each category.

**Figure 5 biomolecules-12-00961-f005:**
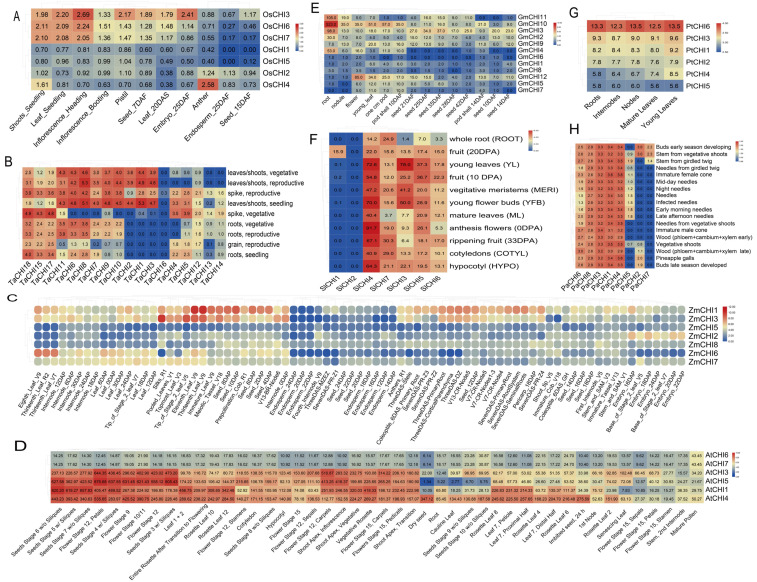
Expression patterns of *CHI* genes in various tissues from eight green plant species, including *O. sativa* (**A**), *T. aestivum* (**B**)*, Z. mays* (**C**), *A. thaliana* (**D**), *G. max* (**E**), *S. lycopersicum* (**F**), *P. trichocarpa* (**G**), and *P. abies* (**H**), respectively. The values in the color scale indicate the relative expression levels.

**Table 1 biomolecules-12-00961-t001:** Statistics of all orthologous groups (OGs) among 15 green plants.

	OG1	OG2	OG3	OG4	OG5	OG6	OG7	OG8	Total
AtCHI	3	1	1	1	1	0	0	0	7
BoCHI	4	1	1	1	2	0	0	0	9
CqCHI	3	1	1	1	1	0	0	0	7
CrCHI	0	2	0	1	0	0	0	0	3
CriCHI	1	3	1	1	1	1	0	0	8
CsCHI	1	1	1	0	1	0	1	0	5
GmCHI	4	2	2	2	2	0	0	0	12
OsCHI	1	2	2	1	1	0	0	0	7
PaCHI	4	0	1	1	2	0	0	0	8
PpCHI	0	1	3	1	1	1	0	0	7
PtCHI	1	2	1	1	1	0	0	0	6
SlCHI	2	1	1	1	1	0	1	0	7
SmCHI	2	2	2	2	0	0	0	0	8
TaCHI	4	6	3	3	2	0	0	0	18
ZmCHI	2	3	1	1	1	0	0	2	10
Total	32	28	21	18	17	2	2	2	122
Tajima’s D	−0.8743	−0.83226	−0.72235	−0.70628	−0.7016	-	-	-	-

OG—orthologous group. Colors from green to red in table represent the number of genes of OGs in individual plants from lower to higher. Os, Zm, Ta, At, Bo, Cq, Cr, Cs, Gm, Pp, Pt, Sl, Sm, and Cri represent *Oryza sativa*, *Zea mays*, *Triticum aestivum, Arabidopsis thaliana*, *Brassica oleracea*, *Chenopodium quinoa*, *Chlamydomonas reinhardtii*, *Cannabis sativa*, *Glycine max*, *Physcomitrella patens*, *Populus trichocarpa*, *Solanum lycopersicum*, *Selaginella moellendorffii* and *Ceratopteris richardii* respectively.

**Table 2 biomolecules-12-00961-t002:** Ka/Ks analysis for the duplicated *CHI* paralogs in green plants.

Duplicated Pairs	Ka	Ks	Ka/Ks	Duplication Type	Purifying Selection
*OsCHI6*-*OsCHI7*	0.0190	0.1560	0.1219	SD	YES
*TaCHI1*-*TaCHI2*	0.0068	0.0960	0.0708	SD	YES
*TaCHI1*-*TaCHI3*	0.0137	0.0857	0.1599	SD	YES
*TaCHI4*-*TaCHI5*	0.0163	0.1209	0.1348	SD	YES
*TaCHI6*-*TaCHI8*	0.0110	0.0176	0.6250	SD	YES
*TaCHI6*-*TaCHI10*	0.0166	0.0544	0.3051	SD	YES
*TaCHI7*-*TaCHI9*	0.0111	0.0913	0.1216	SD	YES
*TaCHI7*-*TaCHI11*	0.0055	0.0721	0.0763	SD	YES
*TaCHI8*-*TaCHI10*	0.0055	0.0737	0.0746	SD	YES
*TaCHI9*-*TaCHI11*	0.0055	0.0531	0.1036	SD	YES
*TaCHI12*-*TaCHI13*	0.0109	0.2043	0.0534	SD	YES
*TaCHI12-TaCHI14*	0.0055	0.1559	0.0353	SD	YES
*TaCHI13-TaCHI14*	0.0165	0.0364	0.4533	SD	YES
*TaCHI15-TaCHI17*	0.0111	0.0174	0.6379	SD	YES
*TaCHI15-TaCHI18*	0.0055	0.0721	0.0763	SD	YES
*TaCHI17-TaCHI18*	0.0055	0.0914	0.0602	SD	YES
*ZmCHI8-ZmCHI5*	0.0740	0.1383	0.5351	SD	YES
*ZmCHI6-ZmCHI1*	0.0453	0.1117	0.4056	SD	YES
*BoCHI1-BoCHI8*	0.1028	0.2994	0.3434	SD	YES
*BoCHI4-BoCHI6*	0.0165	0.5187	0.0318	SD	YES
*BoCHI6-BoCHI7*	0.1409	0.2629	0.5359	TD	YES
*GmCHI9-GmCHI1*	0.0378	0.1303	0.2901	SD	YES
*GmCHI3-GmCHI2*	0.0274	0.0782	0.3504	SD	YES
*GmCHI10-GmCHI4*	0.2124	1.0447	0.2033	SD	YES
*GmCHI7-GmCHI5*	0.0245	0.1113	0.2201	SD	YES
*GmCHI8-GmCHI6*	0.0108	0.1443	0.0748	SD	YES
*GmCHI10-GmCHI11*	0.2204	1.0763	0.2048	TD	YES
*GmCHI11-GmCHI12*	0.3672	1.3982	0.2626	TD	YES
*SmCHI7-SmCHI1*	0.0171	0.0371	0.4609	SD	YES
*SmCHI8-SmCHI2*	0.0043	0.0140	0.3071	SD	YES
*SmCHI5-SmCHI3*	0.0130	0.0138	0.9420	SD	YES
*SmCHI6-SmCHI4*	0.0071	0.0076	0.9342	SD	YES
*CqCHI2-CqCHI3*	0.0638	0.5225	0.1221	TD	YES
*PpCHI2-PpCHI7*	0.0699	0.6405	0.1091	SD	YES
*PtCHI2-PtCHI5*	0.0719	0.2213	0.3249	SD	YES
*SlCHI2-SlCHI3*	0.1714	0.4364	0.3928	TD	YES

SD, mean segmental duplication; TD, mean tandem duplication. Os, Zm, Ta, At, Bo, Cq, Cr, Cs, Gm, Pp, Pt, Sl, Sm, and Cri represent *O. sativa*, *Z. mays*, *T. aestivum, A. thaliana*, *B. oleracea*, *C. quinoa*, *C. reinhardtii*, *C. sativa*, *G. max*, *P. patens*, *P. trichocarpa*, *S. lycopersicum*, *S. moellendorffii* and *C. richardi*, respectively.

## Data Availability

Not applicable.

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
