# Peer review of "Genome-Wide Classification and Evolutionary Analysis Reveal Diverged Patterns of Chalcone Isomerase in Plants"

_biomolecules, 2022, doi:10.3390/biom12070961_

Round 1
Reviewer 1 Report
The manuscript describes a genome-wide classification and evolutionary analysis of the chalcone isomerase gene in plants. They carried out phylogenetic analysis among 15 green plant species from chlorophyte algae to angiosperms to allow the identification of shared and specific subfamilies into the CHI family. The identified CHIs were classified into three subfamilies. This article increases the scientific community's knowledge concerning the evolution and potential function of plant CHI genes. The experiments have been described in detail and the results are discussed adequately. The manuscript is well written. While generally well done, this reviewer thinks this MS could be accepted after minor revision.
I have included below a few concerns.
The resolution of figure 2 and figure 4 is too low. I suggest the authors change the font size of figure4 which is hard to see clearly.
Line 212: there should be a space between N2 and immediately.
Line 286: “these tested vascular species tree during evolution”, delete “tree”.
Line 287: delete “have”.
Line 443: change “are” to “were”.
Line 444: change “are” to “were”.
Line 495: change “This” to “These”.
Line 496: add “a” before “closer”.
Line 588: add “to” before “our previous”.
Line 589: add “there was” before “a clear transcriptional”, add “of” before “CHIs”.
Line 590: add “the” before “same types”.
Line 614: add “patterns” before “among various tissues”.
Line 751: why are “six major subfamilies” here? Is it four?
Line 786: FAP III CHIs?? Is it correct?
Line 837: change “needs” to “need”.
Line 849: change “chlorophytic” to “chlorophyte”.
Line 861: add “of ” to “the molecular characteristics”.
Author Response
We thank the reviewers’ positive notes as well as critical but constructive comments to strengthen our manuscript. Reviewers comments are shown with a black font. Our responses are marked by red font. New additions and changes in the main text are also marked by red font.
Comments:
The manuscript describes a genome-wide classification and evolutionary analysis of the chalcone isomerase gene in plants. They carried out phylogenetic analysis among 15 green plant species from chlorophyte algae to angiosperms to allow the identification of shared and specific subfamilies into the CHI family. The identified CHIs were classified into three subfamilies. This article increases the scientific community's knowledge concerning the evolution and potential function of plant CHI genes. The experiments have been described in detail and the results are discussed adequately. The manuscript is well written. While generally well done, this reviewer thinks this MS could be accepted after minor revision.
I have included below a few concerns.
Point 1: The resolution of figure 2 and figure 4 is too low. I suggest the authors change the font size of figure4 which is hard to see clearly.
Response 1: Thanks very much for your comments. Considering your suggestion, we have increased a higher font size and resolution of the figure 2 and figure 4 for the readability.
Point 2: Minor revision:
Line 212: there should be a space between N2 and immediately.
Line 286: “these tested vascular species tree during evolution”, delete “tree”.
Line 287: delete “have”.
Line 443: change “are” to “were”.
Line 444: change “are” to “were”.
Line 495: change “This” to “These”.
Line 496: add “a” before “closer”.
Line 588: add “to” before “our previous”.
Line 589: add “there was” before “a clear transcriptional”, add “of” before “CHIs”.
Line 590: add “the” before “same types”.
Line 614: add “patterns” before “among various tissues”.
Line 751: why are “six major subfamilies” here? Is it four?
Line 786: FAP III CHIs?? Is it correct?
Line 837: change “needs” to “need”.
Line 849: change “chlorophytic” to “chlorophyte”.
Line 861: add “of ” to “the molecular characteristics”.
Response 2: We are grateful to the reviewer for suggesting these revisions. Each of the errors have been corrected in the main text as suggested. The detail corrections are as following:
A space has been added between “N2” and “immediately”( p. 4 line 212 ).
The two word of “tree” and “have” have been deleted from their corresponding sentence ( p. 7, line 286 and 287 ).
We have separately changed “are” into “were” in the revised manuscript (p.11, line 443 and 444).
“This” has been changed to “These” in the revised manuscript (p.14, line 495).
We have added “a” before “closer” (p.14, line 496).
In accordance with your suggestion, we have corrected these errors, namely, these words of “to”, “there was”, “of”, “the”, and “patterns” have been added (p.16, line 588-590 and line 614).
We are very sorry for this error, and the “six major subfamilies” has been corrected to “four major subfamilies” here (p.20, line 751).
We have changed the typing error of “FAP III CHIs” to “FAP3 CHIs” in the revised manuscript (p.20, line 786).
In the revised manuscript, “needs” and “chlorophytic” have been corrected to “need” and “chlorophyte”, respectively (p.21, line 837 and p.22, line 849)
The word of “of ” have been added before “the molecular characteristics” in the revised manuscript (p.21, line 861).
Thank you very much again for your valuable suggestions for improvement of our paper.

Reviewer 2 Report
I checked your manuscript and described comment below.
This paper does a good study on Phylogenetic relationship and functional diversification of CHI family genes in green plants.
However, there are problems with the following points.
The characters in the figures 1, 2, 3, 4 and 5 are too small to look good. These figures should be larger characters.
I don't think this paper has any major mistakes or grammatical problems.
Author Response
We express our sincere gratitude to your constructive comments to strengthen our manuscript. Our responses (red fonts) follow your comments.
Comments:
I checked your manuscript and described comment below.
This paper does a good study on Phylogenetic relationship and functional diversification of CHI family genes in green plants.
However, there are problems with the following points.
I don't think this paper has any major mistakes or grammatical problems.
Point 1: The characters in the figures 1, 2, 3, 4 and 5 are too small to look good. These figures should be larger characters.
Response 1: Thank you very much for your comments and affirmation on our paper. In accordance with your suggestion, we have reformatted and increased a larger resolution of all figures in the revised manuscript. In additon, a higher resolution version for them is submitted with Figures_Graphics_ Images material.

Reviewer 3 Report
The research article entitled "Genome-wide classification and evolutionary analysis reveal diverged patterns of chalcone isomerase in plants" by Jianyong Wang et al., for the first time provides comprehensive characteristics analysis and proposes a phylogenetic classification of 122 studied CHI genes among fifteen green plants. The introduction provides all the necessary background information and clearly defines the aim of the study. Results are well written, clearly presented, sound and convincing. Materials and methods are adequately presented.
The only concern is the need of some English correction and minor issues with figure 4 and supplementary materials:
1. Figure 4. The current size and resolution makes it unreadable even at the highest zoom levels. It should be either reformatted (for instance, A and B positioned one over another, not side by side) or a higher resolution version should be moved to the supplementary material.
2. Supplementary materials. It would be beneficial to provide the name and description of the material on the same slides.
2.1 Table S1. Imformation → information
2.2 Table S7A. The datail information of cis-acting elements from 122 CHI geens in 15 green plants → The detail information of cis-acting elements from 122 CHI genes in 15 green plants
2.3 Figure S4. What do the colours (blue and red) of the modelled CHIs mean. Please, provide a legend to the figure.
3. Small but numerous issues with the language.
For instance:
Line 12: “However, the phylogenetic dynamics and functional divergence of CHI family genes during the evolutionary path of green plants remain relatively limited investigation”.
Line 70: “Type I CHIs are widely in vascular plants and are involved in the production …”. → Are widely present in … ?
...
Line 604: “… OsCHI3 was relatively highly expression in almost all tested tissues were all highly expression in leaves/shoots and spike at the reproductive stage …”
Line 684: “Notably, no expression of AtCHI2 was observe in any tissues, suggesting AtCHI2 may be ...”
Line 768: “Interestingly, the previous results showed that SmCHI was more higher catalytic activity to naringenin chalcone ...”
Line 817: “… level implying the functional redundant between them …” → redundancy ?
And so on.
I think this article is of a great interest to the readers and should be improved after a minor revision addressing aforementioned Figure 4 and language issues.
Author Response
We express our sincere gratitude for your critical but constructive comments to improve our manuscript. Reviewers comments are shown with a black font. Our responses are marked by red font. New additions and changes in the main text are marked by red font using the “Track Changes” function in MS Word.
Comments:
The research article entitled "Genome-wide classification and evolutionary analysis reveal diverged patterns of chalcone isomerase in plants" by Jianyong Wang et al., for the first time provides comprehensive characteristics analysis and proposes a phylogenetic classification of 122 studied CHI genes among fifteen green plants. The introduction provides all the necessary background information and clearly defines the aim of the study. Results are well written, clearly presented, sound and convincing. Materials and methods are adequately presented.
The only concern is the need of some English correction and minor issues with figure 4 and supplementary materials:
Point 1: Figure 4. The current size and resolution makes it unreadable even at the highest zoom levels. It should be either reformatted (for instance, A and B positioned one over another, not side by side) or a higher resolution version should be moved to the supplementary material.
Response 1: Thanks very much for your comments. Considering your suggestion, we have reformatted to increased a higher font size and resolution of the figure 4 for the readability. In addtion, in order to keep a higher resolution version, we only retain the previous figure 4B in the revised paper, and the previous figures 4A is moved to the supplementary material with a higher resolution.
Point 2: Supplementary materials. It would be beneficial to provide the name and description of the material on the same slides.
2.1 Table S1. Imformation → information
2.2 Table S7A. The datail information of cis-acting elements from 122 CHI geens in 15 green plants → The detail information of cis-acting elements from 122 CHI genes in 15 green plants
2.3 Figure S4. What do the colours (blue and red) of the modelled CHIs mean. Please, provide a legend to the figure.
Response 2: Thank you very much for your very important suggestions on our paper. To address your comments, we have corrected those typing errors and have provided the detail name and description of the figures and tables in the supplemeatary materials. These detail corrections are as following:
The typing errors of “Imformation”, “datail”, and “geens” have been revised to “information”, “detail”, and “genes” in our paper.
The legends are provided in figure S4, namely, blue colors represent α-helices and red colors represent β-stranded sheet. α-helices and β-stranded sheets connected by loops, and ball and stick represent the ligands.
Point 3: Small but numerous issues with the language.
For instance:
Line 12: “However, the phylogenetic dynamics and functional divergence of CHI family genes during the evolutionary path of green plants remain relatively limited investigation”.
Response: “…relatively limitd investigation” has beeen changed to “ …the poorly understood” based on your suggestion.
Line 70: “Type I CHIs are widely in vascular plants and are involved in the production …”. → Are widely present in … ?
Response: We agree with the reviewer’s comment, and the word of “present “ is added between “widely” and “in” to increase the accuracy.
Line 604: “… OsCHI3 was relatively highly expression in almost all tested tissues were all highly expression in leaves/shoots and spike at the reproductive stage …”
Response: We have made correction the previous sentence of “… OsCHI3 was relatively highly expression in almost all tested tissues….were all highly expression in leaves/shoots and spike at the reproductive stage …” to “… OsCHI3 was highly expressed in almost all tested tissues were all highly transcribed in leaves/shoots and spike at the reproductive stage …” according to the Reviewer’s comments.
Line 684: “Notably, no expression of AtCHI2 was observe in any tissues, suggesting AtCHI2 may be ...”
Response: As Reviewer suggested, the sentence of “Notably, no expression of AtCHI2 was observe in any tissues, suggesting AtCHI2 may be ....” in line 684 has been changed to “Notably, no expression of AtCHI2 was observed in any tissues, suggesting AtCHI2 may be ...” in the revised paper.
Line 768: “Interestingly, the previous results showed that SmCHI was more higher catalytic activity to naringenin chalcone ...”
Response: As Reviewer suggested, the sentence of “Interestingly, the previous results showed that SmCHI was more higher catalytic activity to naringenin chalcone ...” in line 768 has been corrected to “Interestingly, according to previous results, SmCHI exhibited more favorable activity to naringenin chalcone which was similar to the type I CHI enzyme characteristic ...” in the revised manuscript.
Line 817: “… level implying the functional redundant between them …” → redundancy ?
And so on.
I think this article is of a great interest to the readers and should be improved after a minor revision addressing aforementioned Figure 4 and language issues.
Response 3: We agree with the reviewer’s suggestion, and the word of “redundant” from the sentence of “… level implying the functional redundant between them …” in line 817 have been revised to “redundancy”.
To address your comments, we have carefully revised the whole manuscript again and again to avoid the similar pboblem again in the new manuscript. We are again special grateful to you for suggesting these revisions to improve our paper.
